# Integration of Low-Cost Digital Tools for Preservation of a Sustainable Agriculture System

Alejandra Serrano-Trujillo *, José Jaime Esqueda-Elizondo [ID] and Laura Jiménez-Beristáin

Facultad de Ciencias Químicas e Ingeniería, Universidad Autónoma de Baja California, Tijuana BC 22427, Mexico; jjesqueda@uabc.edu.mx (J.J.E.-E.); ljimenezb@uabc.edu.mx (L.J.-B.)
* Correspondence: aserrano11@uabc.edu.mx

**Abstract:** This work presents an electronic sensing approach composed of a pair of Physical–Chemical and Imaging modules to preserve an aquaponic system. These modules offer constant measurements of the physical–chemical characteristics within the fish tank and the grow bed, and an indication of the health of the growing plants through image processing techniques. This proposal is implemented in a low-cost computer, receiving measurements from five sensors, including a camera, and processing the signals using open-source libraries and software. Periodic measurements of the temperature, water level, light, and pH within the system are collected and shared to a cloud platform that allows their display in a dashboard, accessible through a web page. The health of the vegetables growing in the system is estimated by analyzing visible and infrared spectra, applying feature extraction, and computing vegetation indices. This work provides a low-cost solution for preserving sustainable urban farming systems, suitable for new farming communities.

**Keywords:** precision agriculture; sustainable agriculture; NDVI; feature extraction; smart farming; support vector machine

## 1. Introduction

The integration of electronic devices and signal processing techniques for solving repetitive or intensive visual tasks has boosted the manufacturing, medical and food industries [1–3]. The technology development seen as a result of computer vision applications has reached agriculture as an area with social impact, where ensuring food security represents a significant challenge given the growth of the global population [4]. Global and local agriculture monitoring has been addressed by remote sensing platforms, active and passive sensors, and applications related to yield prediction, irrigation, and weed detection [5], grouped for precision farming. Smart farming (a term equivalent to precision farming) has emerged as agriculture for ensuring food security [6], supporting ecologically and economically sound agricultural management via site-specific applications. As a result, the use of smart devices and different applications of precision and automatized agriculture has improved the efficiency of agriculture [4]. Moreover, the increase in popularity of the Internet of Things has dramatically benefitted the Raspberry Pi [7], a low-cost computer representing an enabler of technology.

On the other hand, the increasing demand for food production has led to the expansion of protected and soil-less cultivation systems, such as aquaponic and hydroponic systems [8]. These systems yield high productivity in small spaces and represent a solution for agriculture in arid lands, in saline-prone areas, in urban and suburban environments, or wherever the competition for land and water or unfavorable climatic conditions require the adoption of intensive production systems [8]. As with traditional agriculture techniques, soil-less cultivation systems require constant monitoring of visual and physical–chemical characteristics [9], turning interesting the use of smart farming techniques under these conditions. Given their spatial requirements, the implementation of soil-less cultivation

systems may appear easier to monitor and control; however, the interaction of their subsystems, especially when it comes to aquaponics, involves constant measurements from the environment.

Considering the increasing need for supporting sustainable agriculture in developing nations for means of food security, as well as the temperate–arid climate and decreasing water availability in the region of Baja California, Mexico [10], we propose the implementation of a small-scale aquaponics system and its preservation employing smart farming techniques. Aquaponics consists of two parts that combine aquaculture and hydroponic systems [8]. Therefore, the critical variables for preservation are related to the quality of the fish tank's water, the sunlight over the system, and the health of the growing plants, which in this proposal are cherry tomatoes. The health of the fish is indirectly monitored through the pH sensor (for an optimal pH range) and measuring the water level from the tank (to detect an obstruction or overflow on time). Regarding the health of the plants, a Normalized Difference Vegetation Index (NDVI) analysis may provide early detection of plant stress [11], supporting this smart farming approach with information beyond the visible spectrum [12].

The lack of constant attention to any of the variables from an aquaponic system could rapidly lead to the death of fishes in the tank, to an eventual nutrient deficiency or to the presence of pests [13]. These consequences highlight the importance of monitoring an aquaponic system even when implemented by expert users. All these characteristics can be monitored through different sensors, gathered and processed through a computer, and transmitted and displayed using a cloud application. The temperature, light, ultrasonic, and pH sensors and the Raspberry Pi and its NoIR camera module represent low-cost elements for an integral solution to preserve sustainable agriculture systems. Here, signal and image processing techniques, exploiting open-source software and libraries, become part of the solution for delivering the essential data so the user can implement changes over the system when required.

This document is organized by presenting in Section 2 the aquaponic system, the Physical–chemical and Imaging modules, the techniques applied over the infrared and visible images, and the implementation of a support vector machine (SVM) for classification. Section 3 presents the signals collected from both modules that are organized and analyzed, leading to a discussion regarding the behavior of the physical–chemical measurements over time and the performance of the estimated features. Finally, in Section 4, our conclusions regarding the current performance of the proposed modules and future work while maintaining a low-cost approach are presented.

## 2. Materials and Methods

This proposal integrates a sustainable agriculture system with low-cost data acquisition devices, open-source software, and techniques for signal processing and data display. This section describes the aquaponic system, the pair of modules proposed for imaging and for sensing physical–chemical parameters, and the image processing techniques involved.

### 2.1. Aquaponic System

Aquaponics combines aquaculture and hydroponics in a continuously recirculating unit composed of a fish tank and plant grow beds. In this system, an air pump allows the flow of culture water containing the metabolic wastes of fish from the fish tank to a mechanical filter that captures solid wastes and a biofilter that oxidizes ammonia to nitrate. Then, this water travels through plant grow beds, where plants uptake the nutrients, and finally, it is released back to the fish tank through a siphon [8]. When this system is balanced correctly, all the organisms work together to create a healthy growing environment for one another, leading in this approach to the need for constant monitoring through electronic sensors. The main elements of the interaction described for an aquaponic system are presented in Figure 1.

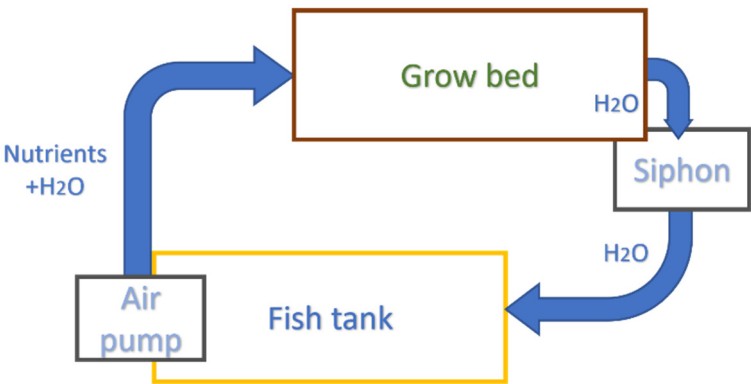

**Figure 1.** Diagram of the components of an aquaponic system.

The grow bed illustrated in Figure 1 is meant to contain cherry tomatoes, which are fruiting vegetables that perform exceptionally well in aquaponics despite having high nutrient demands [8]. It is then proposed to locate the electronic devices for monitoring this system in the two core elements from Figure 1, the fish tank and the grow bed.

*2.2. Physical–Chemical Sensing Module*

The module for sensing physical and chemical characteristics within the aquaponic system, including the imaging system, is depicted in Figure 2. A pH meter composed of the PH-4502C board and a BNC probe is used, locating the pH probe inside the fish tank. A digital temperature probe using the DS18B20 sensor is also located inside the fish tank, and an ambient light sensor, the MAX44009 circuit, is placed over the plant grow beds. The ultrasonic sensor HC-SR04 is located inside a lid covering the fish tank, facing down to detect a significant change in water level. These sensors are connected to a Raspberry Pi 3, model B, through the General-Purpose Input/Output (GPIO) port. These devices are selected given their power consumption, working temperature, measuring range, and cost according to the environment's expected conditions. To interpret the data received from the pH, ultrasonic and ambient light sensors, the signals are processed through the ADS1115, an analog to digital converter. Each sensor may require a specific communications protocol, the I2C or 1-Wire when connected to the GPIO (General-Purpose Input/Output) pins.

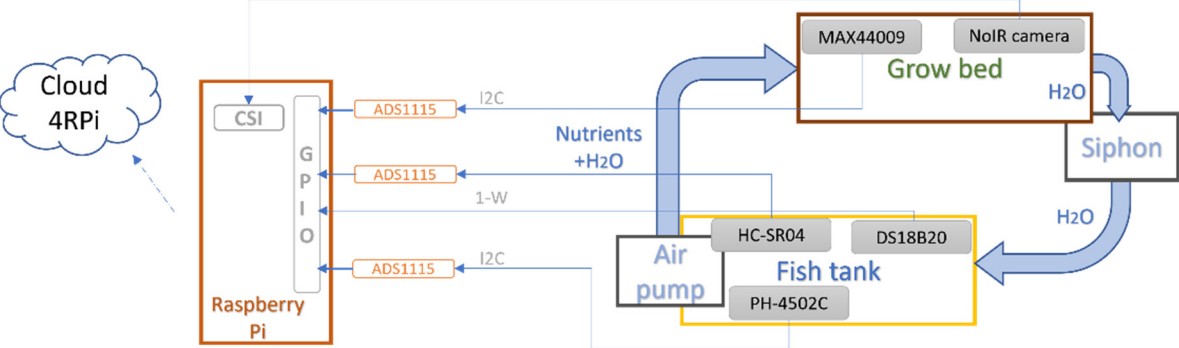

**Figure 2.** Diagram of electronic sensing modules over the aquaponic system.

The data collected from the physical–chemical Sensing Module is sent to Cloud4RPi, a cloud platform that allows the connection and control of a Raspberry Pi, using customizable dashboards that are accessible through mobile devices [14]. A user is created for this platform, and four widgets are selected for displaying the data received from the sensors connected to the Raspberry Pi. According to the theory and to our experience with aquaponic systems, an email alert is configured for the pH when measured outside the range of 6.8 to 7.8 [8]. According to the selected platform, data are sent at a rate that allows

the user to visualize the four sensors at no financial cost. In this case, a measurement is shared every hour for the temperature and light sensors, every two hours for the ultrasonic sensor, and every 20 min for the pH probe. Considering a loss of internet connection may occur, Browser for SQLite is set for recording the data read from the sensors in the Raspberry Pi. This open-source tool allows the organization and manipulation of databases, which is helpful in this approach to export tables and plot graphs of the behavior of the physical–chemical parameters within the system over different time intervals. The interaction of the elements of the imaging module, illustrated as well in Figure 2, is described in the following section.

### 2.3. Imaging Module

The imaging module comprises a NoIR camera, a Raspberry Pi compatible camera with no IR filter that allows registering both visible and near-infrared spectra, connected directly to the CSI (Camera Serial Interface) port. A short-pass and a long-pass filter are part of the imaging module, with a cut-on wavelength at 750 nm. These filters allow capturing individually, through the camera, the visible and near IR spectra. This information is useful for computing the NDVI, which uses measured reflectance values in the visible and NIR (near-infrared) regions to provide valuable information related to the health of plants, estimating crop growth, vigor, and photosynthesis [15].

The mathematical expression for computing NDVI is presented in Equation (1)

$$\text{NDVI} = \frac{R_{NIR} - R_{red}}{R_{NIR} + R_{red}}, \tag{1}$$

$R_{NIR}$ represents the image obtained using the long-pass filter (allowing near-infrared light only), and $R_{red}$ represents the image's red channel obtained using the short-pass filter (allowing visible light only). The value of NDVI ranges from $-1$ to $1$. Positive values indicate increasing greenness, while negative values indicate non-vegetated surfaces, such as bare soil/land or water [15]. Moreover, working specifically with tomato plants, the vegetation indices RVI (Ratio Vegetation Index) and NDGI (Normalized Difference Greenness Index) are presented in Equations (2) and (3) and are helpful for segmentation. RVI provides an image with increased contrast on vegetation areas, while NDGI provides an easy solution for locating the tomatoes once they start turning red.

$$\text{RVI} = \frac{R_{NIR}}{R_{red}}, \tag{2}$$

$$\text{NDGI} = \frac{R_{green} - R_{red}}{R_{green} + R_{red}}. \tag{3}$$

The elements of the imaging module are presented in Figure 3.

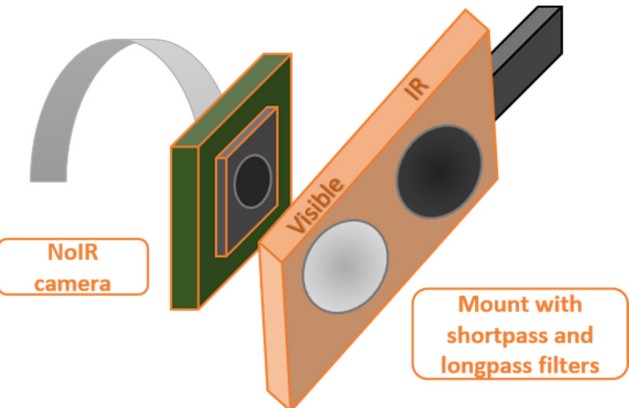

**Figure 3.** The imaging module comprises a NoIR camera, a long-pass, and a short-pass filter.

Both $R_{NIR}$ and $R_{red}$ images are captured using a plastic mount that allows a fast location of each filter in front of the camera sensor. NDVI is computed for these images, and the characteristics found are analyzed through feature extraction and other image processing techniques, as described in the following section.

### 2.4. Image Processing over Visible and Near-Infrared Spectra

Images of the tomatoes are captured 20 to 30 cm away from the edge of the grow bed. The visible and near-infrared spectra filters capture these images with the NoIR camera by pairs. In Figure 4, the image processing techniques applied to estimate the plants' health are presented.

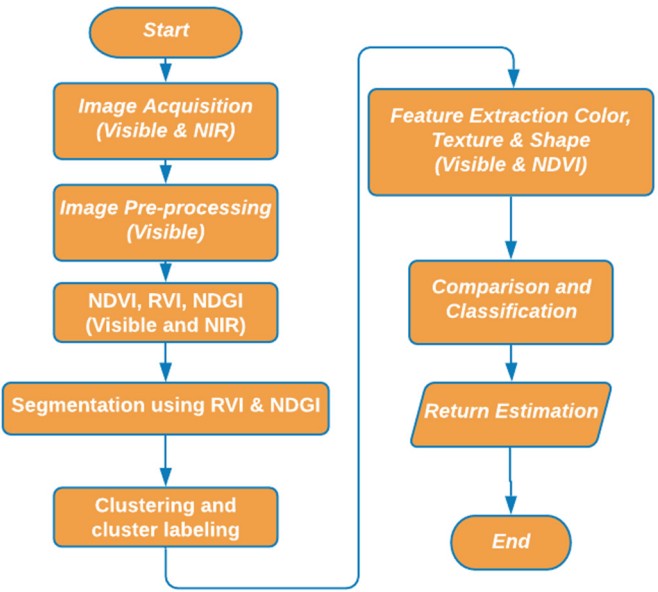

**Figure 4.** Image processing techniques applied over each set of spectral images.

As depicted in Figure 4, the pair of images are labeled as *Visible* and *NIR*, given their spectral characteristics. Both images are pre-processed by applying a bilateral filter for edge-preserving smoothing. Then, the vegetation indices NDVI, RVI, and NDGI are computed, resulting in three two-dimensional arrays. The latter two allow the visible image's binarization for locating tomatoes that are turning red. The groups of pixels detected are validated by computing their total area, and clustering is performed for image segmentation. Color, shape, and texture features and NDVI are computed for each object detected in the segmentation stage. The extracted shape and texture features are classified and compared to a support vector machine (SVM) training set. All these routines are applied to the program in Python, using libraries from OpenCV.

Additionally, the user can capture images of the tomato leaves to estimate NDVI, computing this index directly.

### 2.5. Support Vector Machine

Support vector machines have been widely used in computer vision [16]. In this work, we use an SVM for classification by importing a training set composed of 22 images from the grow bed of our aquaponic system. The set includes examples of *round* and *occluded* tomatoes for the shape feature. Most of the set is composed of *smooth* and *split* tomatoes for the texture feature, with only a few examples of rotten tomatoes. Texture and shape features are extracted by computing Haralick features [17] and Hu moments [18], respectively. Then, a linear SVM classifier is created and the training features and labels, previously computed, are fitted. Once this classifier is created, the comparison and classification described at the end of Section 2.4 can be implemented.

## 3. Results and Discussion

The Imaging and physical–chemical Sensing Modules are proposed in this paperwork as an aid in the preservation of our aquaponic system. In this section, we present the results regarding the information offered by each module, collected from May to October 2021.

### 3.1. Physical–Chemical Sensing Module

The module presented in Figure 2 was implemented over the aquaponic system. Measurements were periodically collected during a whole week, representing one testing cycle. The pH probe was removed from the tank and cleaned after each cycle to maintain its performance. The rest of the sensors did not require a maintenance procedure. In Figure 5, the behavior of the four registered variables within different testing cycles is presented. Figure 5a presents the hourly average temperature and illumination over the system during a pair of summer and autumn weeks. The temperature scale, in Celsius degrees, is seen on the left side, while the scale of ambient light is indicated on the right side of Figure 5a in Lux. It is easy to identify the behavior related to each season, where both temperature and illuminance decreased from summer to autumn, as expected given our geographic location. The bars in yellow and gray colors represent the illumination during summer and autumn, respectively. The orange and blue lines represent the average temperature registered during summer and autumn. Regardless of the decrease in temperature seen between seasons, the optimal growth of cherry tomatoes (18.5–26.5 °C) [19] can be achieved during any of these seasons. In this case, additional requirements of the aquaponic system can be estimated through the imaging module.

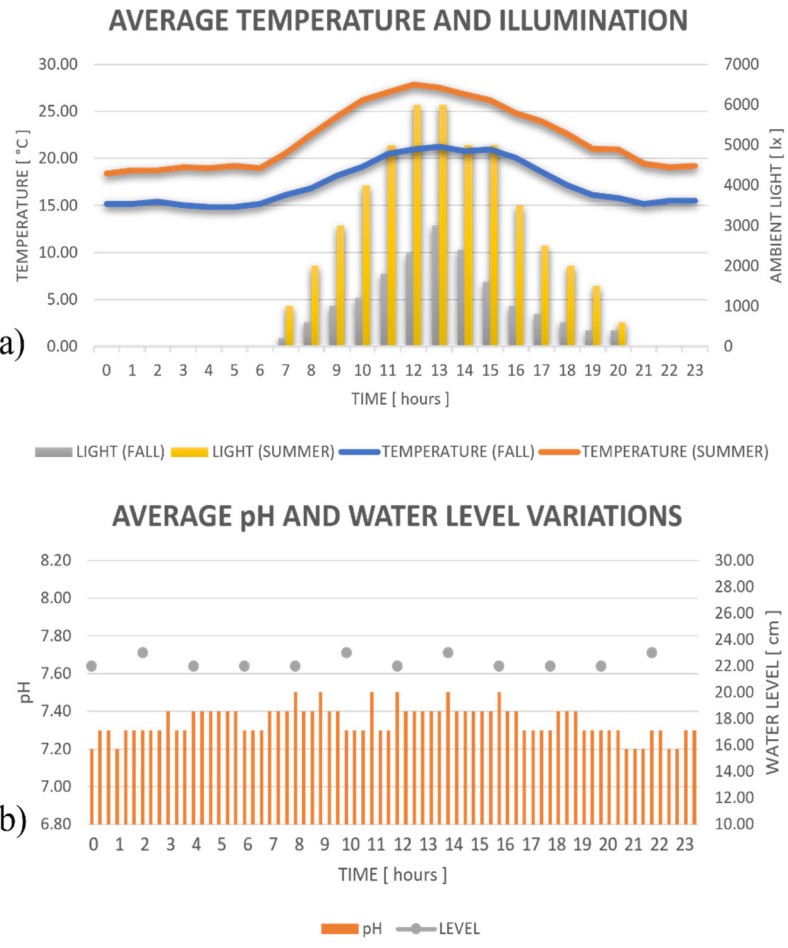

**Figure 5.** Physical–chemical Sensing Module: (**a**) average temperature and illuminance, summer vs. autumn week, (**b**) average pH and water level readings within a week.

Figure 5b presents the pH and water level variations during a week; the scales of these variables are shown on the left and right sides, respectively. This graph did not require a comparison between seasons since no significant differences were found. A water-level reading is performed every two hours, while three pH readings are collected every hour. Water level, represented in gray dots, indicates the distance between the top of the water from the tank and the lid. According to the tank's dimensions, an absolute variation of 2 cm indicates that 20 L of water are circulating between the fish tank and the grow bed. A greater level variation would require attention, indicating that something is obstructing the recirculation process, explicitly checking the air pump and the siphon. Therefore, a distance between 20 to 26 cm is set as acceptable in the Cloud4RPi dashboard. In Figure 5b, pH, represented in orange bars, varies from 7.2 to 7.5. Since these measurements are within the 6.8 to 7.8 interval, no additional attention is required.

In Figure 6, we present an example of the dashboard, accessed through a mobile device, showing the status of the variables according to previously specified safe ranges.

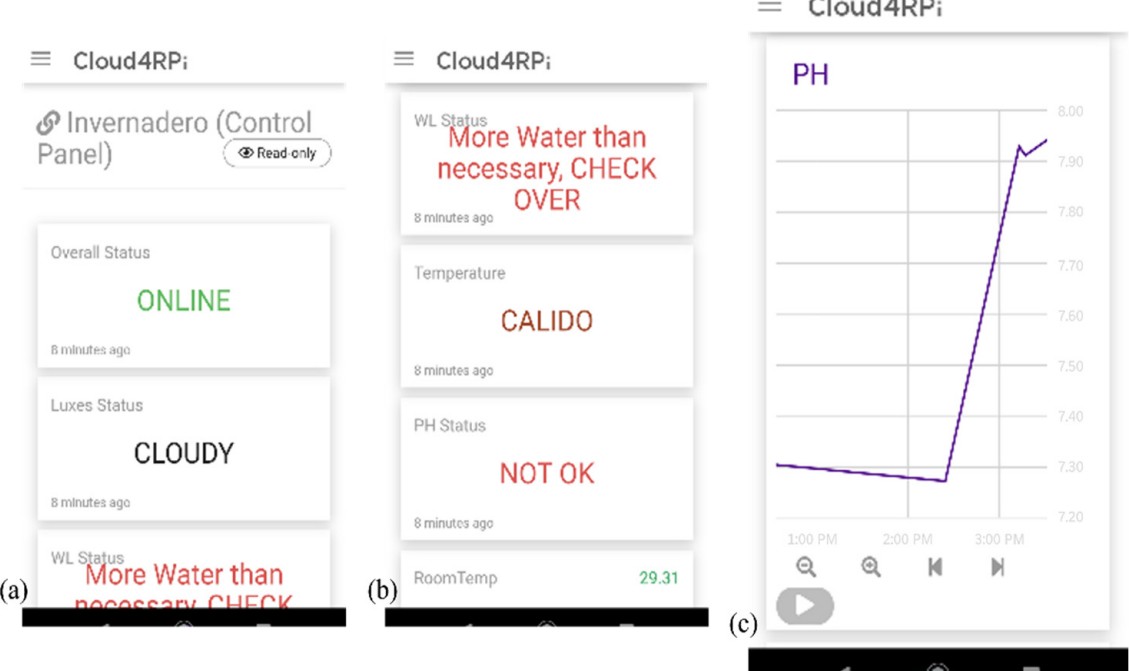

**Figure 6.** Mobile access to the control panel of Cloud4RPi: General Status of (**a**) illuminance, (**b**) water level, temperature, and pH. (**c**) The behavior of pH during the last hours.

The examples presented in Figure 6a,b briefly indicate the status of the aquaponic system, which in this case, is detecting a pH value outside of the acceptable range. Figure 6c shows the behavior of this variable during the last hours, where the user can identify the specific pH measured inside the fish tank to add an appropriate substance; in this case, phosphoric acid would be suggested.

These readings and the plant health estimation obtained from monitoring the grow bed allow easier preservation of an aquaponic system.

### 3.2. Imaging Module

The location of the tomatoes as well as their features and health are offered as a result of image processing, as presented in Figure 7.

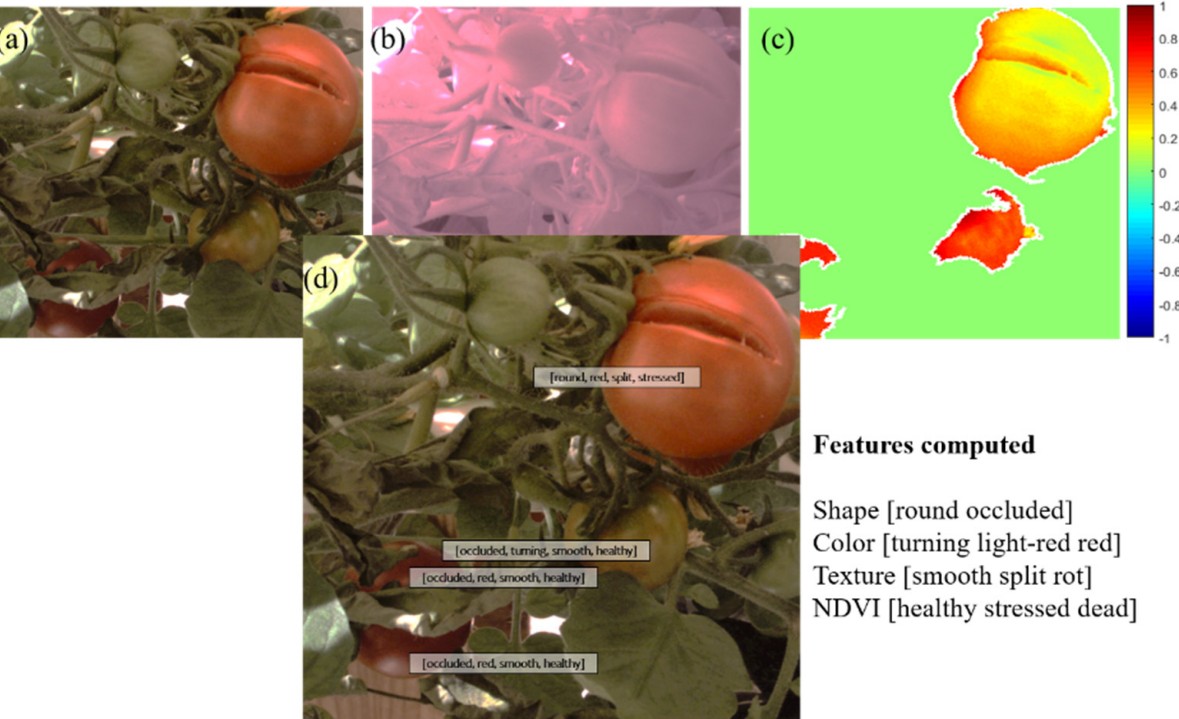

**Figure 7.** Processing of an image from the grow bed (**a**) visible spectrum, (**b**) near-infrared spectrum, (**c**) NDVI of segmented tomatoes, and (**d**) estimation of the features of the located tomatoes.

Figure 7a,b shows a pair of images in the visible and near-infrared spectra from the same location within the grow bed. In these images, four tomatoes can be discriminated in different ripening stages. According to their color, cherry tomatoes can be classified into six different ripening stages [19]. In this approach, we recognize three stages *turning*, *light-red* and *red*. In Figure 7c, the located tomatoes are segmented and displayed in a color scale related to their NDVI, to be later classified as *healthy* (NDVI > 0.66), *stressed* (0.33 < NDVI < 0.66), or *dead* (NDVI < 0.33). Considering the location segments' shape, these are classified as *round* or *occluded* since there may be leaves covering part of them. The texture is also analyzed to identify if the surface is regular (classified as *smooth*), if it is *split* or belongs to a rotten tomato, labeled in the algorithm as *rot*. Finally, in Figure 7d, the original image in the visible spectrum is presented with labels indicating shape, color, texture, and NDVI classification. In this case, three tomatoes were located since the algorithm uses RVI and NDGI to search for red objects within the image, ignoring the green tomato. The estimated features allow a quick conclusion regarding the plant's needs, paying attention in this case to the *split* tomato since the other segments located are classified as *healthy*. On the other hand, the leaves of the tomato plant are processed by capturing an image isolating them from the growing tomatoes. In this case, RVI allows easy segmentation of the plants concerning other objects surrounding them.

Figure 8 presents the NDVI computed for tomato leaves under intended humid conditions, where the color scale on the right allows demonstration of health deterioration occurring from Figure 8a to Figure 8b,c. In Figure 8a, the leaves are estimated as *healthy*, while in Figure 8b,c, captured during the two following weeks, the leaves turn *stressed*, given a notorious decrease of NDVI. This monitoring allows detecting, through image processing, the effects of unattended conditions over the aquaponic system.

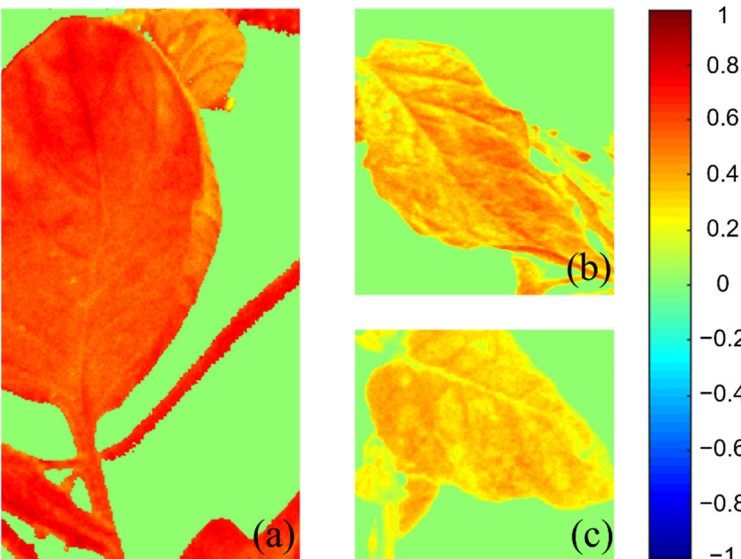

**Figure 8.** NDVI of tomato leaves: (**a**) under healthy conditions, (**b**) after one week under humid conditions, and (**c**) after two weeks under humid conditions.

Considering that it takes around five weeks for the tomato plant to start to flower, fifteen weeks were monitored through image processing, capturing six images around the grow bed every week. Table 1 presents the classification precision regarding the features of shape and texture, computed for 90 images from the grow bed. Accuracy was also computed for both features. These metrics are computed as

$$Precision = \frac{True_{positive}}{True_{positive} + False_{positive}}, \tag{4}$$

$$Accuracy = \frac{True_{positive} + True_{negative}}{True_{positive} + True_{negative} + False_{positive} + False_{negative}}. \tag{5}$$

**Table 1.** The precision of detection of the Shape and Texture Features.

| Shape | | Texture | |
|---|---|---|---|
| *Round* | 0.85 | *Smooth* | 0.95 |
| *Occluded* | 0.94 | *Split* | 0.88 |
| | | *Rot* | 0.66 |

For the *round* feature, for example, *True positive* represents the *round* tomatoes correctly located over the images, *True negative* represents the tomatoes correctly classified as *occluded*, *False positive* is the number of tomatoes classified as *round but* were *occluded*. *False negative* represents the tomatoes classified as *occluded but* were *round*.

From Table 1, *round* tomatoes present lower precision than *occluded* tomatoes. Some clustered tomatoes were considered one object when the color feature led to the same color label. For the texture feature, the lowest precision was reported for rotten tomatoes. There were only a few tomatoes with this texture for training and testing. Some of the rotten cherry tomatoes were classified with a *split* texture when testing for the texture feature.

Conversely, tomatoes with *smooth* and *split* textures were classified with higher precision. The accuracy regarding the shape feature is computed as 89.6% and for texture is 94%. These results mean that most predictions of True positive and True negative categories are correct. In this case, the precision of the texture features labeled as *rot* and *split* requires

attention. This metric could be improved by working with a more extensive set of training images, decreasing the False positives for both categories.

## 4. Conclusions

A smart farming approach for a small-scale aquaponic system was implemented using low-cost tools and open-source software, promoting the preservation of sustainable systems to address food insecurity within communities.

A pair of electronic sensing modules monitor the key parameters that allow an appropriate balance between the fish tank and the grow bed from the aquaponic system. A periodic reading of the physical–chemical parameters and their display through a cloud-based application lets the user know whether the system requires attention. The imaging module estimates the visual features that the user may not recognize on time, offering early detection of nutrient deficiencies and plant diseases. In this work, we were able to detect plant deterioration due to the appearance of a fungus.

The performance of the proposed techniques for classifying shape and texture has been sufficient for this stage. However, a wider training set should improve these results. Moreover, introducing a support vector machine for initially locating the growing tomatoes replacing the approach based on color should let the user know the features and NDVI of all the objects located within the spectral images.

Finally, the spectral analysis implemented in this paper can be enhanced by exploring the interaction of polarized light with the vegetables in the grow bed, maintaining a non-destructive approach to preserving the sustainable system.

**Author Contributions:** Conceptualization, A.S.-T.; methodology, A.S.-T. and J.J.E.-E.; validation, L.J.-B. and J.J.E.-E.; formal analysis, A.S.-T. and J.J.E.-E.; investigation, A.S.-T., J.J.E.-E. and L.J.-B.; resources, A.S.-T. and J.J.E.-E.; data curation A.S.-T. and J.J.E.-E.; writing—original draft preparation, A.S.-T.; writing—review and editing, A.S.-T., J.J.E.-E. and L.J.-B.; visualization, A.S.-T. and J.J.E.-E.; supervision, A.S.-T.; project administration, A.S.-T. All authors have read and agreed to the published version of the manuscript.

**Funding:** This research was funded by Programa para el Desarrollo Profesional Docente (PRODEP, 511-6/2019.-13972).

**Institutional Review Board Statement:** Not applicable.

**Informed Consent Statement:** Not applicable.

**Acknowledgments:** We wish to thank Bruno Jurado Espinoza and Uriel Flores Meza for their technical support during the experiments around the aquaponic system.

**Conflicts of Interest:** The authors declare no conflict of interest.

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
