# Peer review of "Integration of Low-Cost Digital Tools for Preservation of a Sustainable Agriculture System"

_electronics, doi:10.3390/electronics11060964_

Round 1

Reviewer 1 Report

There are some errors. Row 20: Com-munities, row 229 acidwould, row 276 the caption of Table 1 under the Table, row 307 the phrase is not completed. What about the monitoring of fish health?

Author Response

Response to Reviewer 1 Comments

1. There are some errors. Row 20: Com-munities, row 229 acidwould,

Changed to “communities” in line 19 now

Changed to “acid would” in line 239 (added text as a response to reviewer # 2, so it is no longer in line 229)

2. The caption of Table 1 under the Table

For this comment I downloaded the template of this journal and it shows the caption of Table above it, so I left it above the Table, in line 284.

3. Row 307 the phrase is not completed

We completed the last paragraph in lines 316 – 318

4. What about the monitoring of fish health?

Added a quick explanation in lines 57-59

Thanks for the corrections and suggestions,

Reviewer 2 Report

Accepted after minor revision:

  • add "smart farming" and "support vector machine" to keywords.
  • add detailed description of "support vector machine" (SVM) plus at least one source (suggested: https://www.sciencedirect.com/science/article/pii/B9780128157398000067) in new "Section 2.5 Support Vector Machine" to complete 2. Materials and Methods.
  • Finish or delete last paragraph in Section 5 (conclusions).

Author Response

Response to Reviewer 2 Comments

1. add "smart farming" and "support vector machine" to keywords.

Included in lines 20-21

2. add detailed description of "support vector machine" (SVM) plus at least one source (suggested: https://www.sciencedirect.com/science/article/pii/B9780128157398000067) in new "Section 2.5 Support Vector Machine" to complete 2. Materials and Methods.

Included section 2.5 in lines 183-192, added references to SVM and the two feature extraction techniques used for creating the classifier

3. Finish or delete last paragraph in Section 5 (conclusions).

We completed the last paragraph in lines 316 – 318

Thanks for the corrections and suggestions,